# Characterization of Steroid Metabolic Pathways in Established Human and Mouse Cell Models

**DOI:** 10.3390/ijms26199721

**Published:** 2025-10-06

**Authors:** Therina du Toit, Michael Groessl, Emanuele Pignatti, Amanda C. Swart, Christa E. Flück

**Affiliations:** 1Department of BioMedical Research, University of Bern, 3008 Bern, Switzerland; 2Division of Pediatric Endocrinology, Diabetology and Metabolism, Department of Pediatrics, Bern University Hospital, University of Bern, 3010 Bern, Switzerland; 3Department of Biochemistry, Stellenbosch University, Stellenbosch 7600, South Africa; 4Department of Nephrology and Hypertension, Inselspital, Bern University Hospital, University of Bern, 3010 Bern, Switzerland; 5Department of Chemistry and Polymer Science, Stellenbosch University, Stellenbosch 7600, South Africa

**Keywords:** targeted steroid metabolomics, adrenal androgens, placenta, ovary, testis, liquid chromatography-mass spectrometry (LC-MS), 11-oxygenated androgens, cytochrome P450 (CYP) enzymes

## Abstract

Immortalized adrenal, placental and gonadal cell models are often termed steroidogenic based on steroid hormone production and steroidogenic enzymes. Profiling of ‘classic’ steroid metabolites is common; however, downstream untargeted metabolites remain unidentified. This study characterized steroidogenesis in human adrenal H295R and H295A; placental BeWo and JEG-3; mouse Leydig MA-10; and mouse adrenal Y-1 and OS-3 cells. Steroids were determined under basal, stimulated and serum-free conditions using liquid chromatography–mass spectrometry. This study identified distinct differences in mineralocorticoid and glucocorticoid production in the two human adrenal models and between the human and mouse adrenal models; unconventional hydroxylated progesterone steroid metabolites in all models which were most abundant in MA-10 cells; glucocorticoids and abundant classical androgens in MA-10 cells; 11-oxy androgens in H295R, H295A and MA-10 cells; comparable levels of the classical androgens in H295R and MA-10 cells, while 11-oxy androgen were more abundant in H295R and H295A cells; and high pregnenolone and progesterone in placental models with limited hydroxylated progesterone metabolites. Our detailed protocols and comprehensive steroid profiles provide an invaluable guide to researchers for in vitro investigations into steroidogenesis.

## 1. Introduction

Immortalized cell models are well-established as invaluable assets in facilitating our understanding of human development, diseases and disorders. In the laboratory setting, cell models circumvent the need for living organisms, thus allowing the investigation of physiological processes and molecular mechanisms ex vivo. Steroidogenic cell models, in particular, have proved essential in elucidating steroid metabolic pathways that include not only the precursor steroids and end-products, but also intermediate metabolites [1,2,3]. In human steroidogenesis, cholesterol is the precursor molecule to all steroid hormones, and cytochrome P450s, hydroxysteroid dehydrogenases (HSDs) and reductases sequentially convert cholesterol into progestogens, corticosteroids, androgens and their downstream metabolites [4]. These steroidogenic enzymes are endogenously expressed in steroidogenic cell models, permitting de novo steroid biosynthesis (Figure 1).

Advanced analytical technologies such as liquid chromatography–mass spectrometry (LC-MS), having become more accessible, have enabled the profiling of multiple steroid metabolites in single chromatographic steps [5,6]. Comprehensive profiles have, to date, characterized pathways that include the alternative steroid pathway, backdoor pathway and the 11-oxy androgen and 11-oxy progestogen pathways [7,8]. The in vitro tracking and identification of multiple steroid metabolites in steroidogenic pathways have shown steroidogenesis to be far more complex than generally accepted. The quantification of substrates, end-products and intermediate metabolites within pathways (Figure 2) not only highlights relevant steroidogenic enzymes but also informs on steroid fluxes within pathways. Downstream steroid metabolite profiles additionally pinpoint subtle differences, enabling interpretations of modulated pathways in the presence of enzyme inhibitors or activators, natural products, pharmacological drugs and the expression of mutated enzymes [9,10,11].

Steroidogenesis has been extensively studied in established steroidogenic cell models for decades, and in this regard standardized laboratory practices are critical to successful outcomes of investigations into novel steroidogenic pathways and activities. Although protocols are easily accessible online and available from the supplier, these generally describe optimal culturing conditions for growth and maintenance, crucial to conserving the characteristic phenotype and genotype of the cell model. However, experimental conditions do deviate from optimal conditions when treating cells with starvation/serum-free media, phenol red-free media and stimulation aimed at modulating inherent cellular processes [3,12,13]. In addition, exposing cells to inhibitors or activators of specific molecular processes will affect cell model characteristics. Modulated steroid profiles induced by experimental conditions which influence de novo steroidogenesis have, to date, not been established for steroidogenic cell models. Although steroidogenesis has been investigated in a number of cell models, including adrenal H295R cells and mouse Leydig tumor cells [14,15,16], studies generally focus on specific hormones only, and do not include the full array of potential metabolites. Since current validated analytical methods permit the analysis of comprehensive steroid profiles, it has become imperative to characterize steroidogenic pathways in routinely used cell models.

This study therefore investigated steroid profiles in cell models under optimal and modulatory experimental conditions. Cell models included in the study were: human adrenal (H295R and H295A); placental (HTR-8/SVneo, BeWo and JEG-3); ovary (KGN and OVCAR-3); mouse Leydig (MA-10); and mouse adrenal (Y-1 and OS-3) cells. Steroid metabolites were quantified under basal, stimulated and serum-free media conditions. Signature steroid profiles were established using a targeted and untargeted LC-MS approach. This study provides a current perspective on steroid biosynthesis and metabolism in routinely used cell models aimed at supporting the interpretation of steroid metabolic assays.

## 2. Results

Fifty precursor, intermediate and end-product steroid metabolites were included in our LC-MS method which enabled the analysis of the endogenous catalytic activities of the three major classes of steroidogenic enzymes: cytochrome P450s, HSDs and reductases (depicted in Figure 1 and Figure 2). Steroidogenesis was assessed in human and mouse cell models after a 24 h incubation period in either complete growth media or serum-free media. Additional experimental treatments were also conducted. Cells were stimulated with 8-bromoadenosine 3′,5′-cyclic monophosphate (8BrcAMP) to upregulate steroid metabolism by activating cAMP-dependent protein kinases; adrenal cells with metformin to measure the perturbation of steroid biosynthesis by modulating the regulation of steroidogenic enzymes, including 3βHSD [17].

### 2.1. Adrenal Cell Models

#### 2.1.1. Human H295R Cells

All H295R cells remained steroidogenic when cultured in serum-free media. In complete growth media, cells produced mineralocorticoid precursors (mainly 11-deoxycorticosterone and corticosterone), glucocorticoids (mainly 11-deoxycortisol and cortisol) and androgens (mainly dehydroepiandrosterone sulfate [DHEA-S] and androstenedione) (Figure 3a). Serum-free conditions, however, reduced the total steroid output by ~2.5-fold (Figure 3b and Appendix A). 11-Deoxycorticosterone and corticosterone were reduced to trace amounts, a marked decrease in the production of 11-deoxycortisol (~3-fold) and cortisol (~7-fold) was measured, while increased DHEA-S was negligible. Aldosterone, 21-deoxycortisol and cortisone were negligible in H295R cells under the tested conditions.

Stimulation with 8BrcAMP increased steroid production considerably in both complete growth media (~4.4-fold) (Figure 3a) and serum-free media (~3.8-fold) (Figure 3b), while also altering steroid profiles markedly. In complete growth media, the progestogens (pregnenolone and 17α-hydroxyprogesterone [17OH-progesterone]) increased, together with the mineralocorticoids (*p* ≤ 0.01), and the glucocorticoid, 11-deoxycortisol (*p* < 0.0001). Androgens increased considerably, most prominently DHEA-S (*p* ≤ 0.01) and dehydroepiandrosterone (DHEA) (*p* < 0.0001), with significant increases also in testosterone, androsterone and dihydrotestosterone production. In serum-free media, 8BrcAMP restored the production of 11-deoxycorticosterone (*p* ≤ 0.001), but levels remained more than 19-fold lower than stimulated levels in complete growth media. Glucocorticoids were stimulated and production increased ~4-fold, but markedly lower than stimulated production in complete growth media. Androgen production was also stimulated with the increase in DHEA (*p* ≤ 0.01) and dihydrotestosterone most prominently, with androstenedione levels marginally lower than DHEA-S. Although their production was lower than in complete growth media, androstenedione production was greater than that of DHEA-S in complete growth media upon stimulation (~4.7-fold).

The untargeted analysis of steroid production in H295R cells in both complete (Figure 3c) and serum-free media measured additional hydroxylated metabolites of progesterone, androstenedione and testosterone. These included 11α-hydroxyprogesterone (11αOH-progesterone), 11β-hydroxyprogesterone (11βOH-progesterone), 16α-hydroxyprogesterone (16OH-progesterone), 11β-hydroxyandrostenedione (11OH-androstenedione) and 11β-hydroxytestosterone (11OH-testosterone). Downstream metabolites detected in the untargeted analysis indicated catalytic activities of steroid-5α-reductases (SRD5A), 11βHSD, 17βHSD, 20α-hydroxysteroid dehydrogenase (AKRIC1) and 3α-hydroxysteroid dehydrogenase (AKR1C2). The 5α-reduced products included, 5α-androstanedione and 11β-hydroxy-5α-androstanedione (11OH-5α-androstanedione) and 11-ketodihydroprogesterne (11K-dihydroprogesterone). The catalytic activity of 11βHSD type 2 (11βHSD2) was evident in the production of 11-ketoandrostenedione (11K-androstenedione), which was stimulated upon 8BrcAMP treatment (*p* < 0.0001), and negligible 11-ketotestosterone (11K-testosterone) levels. AKRIC1 activity was evident in the production of progesterone and 17OH-progesterone metabolites reduced at C20. Although 8BrcAMP stimulation did not change the steroid profile, metabolites increased upon stimulation, with 6β-hydroxyprogesterone (6βOH-progesterone), 11αOH-progesterone, 20α-hydroxyprogesterone (20αOH-progesterone), 17α,20α-dihydroxyprogesterone (17,20-diOH-progesterone), 11K-androstenedione and 11OH-5α-androstanedione increasing considerably. Under serum-free conditions in the presence of metformin, 17OH-progesterone levels increased ~4-fold (*p* ≤ 0.001); DHEA, ~4-fold; DHEA-S ~2-fold; testosterone, ~2.8-fold; and the ratio of androstenedione/DHEA, denoting 3βHSD type 2 (3βHSD2) activity, decreased ~3-fold (Appendix A).

#### 2.1.2. Human H295A Cells

The H295A cell model, a variant of the H295R cell model, also remained steroidogenic when cultured in serum-free media and produced more mineralocorticoids compared to androgens. H295A cells produced mainly corticosterone, 11-deoxycortisol and cortisol (Figure 4a,b), while 21-deoxycortisol, DHEA, testosterone and its downstream metabolites were end-products negligibly produced in H295A cells under the tested conditions. Serum-free conditions reduced the total steroid output by ~4.5-fold (Figure 4b and Appendix A). 11-Deoxycorticosterone, corticosterone and cortisol were reduced to trace amounts, with a marked decrease in the production of cortisone (~2-fold) and DHEA-S (~4.5-fold).

Stimulation with 8BrcAMP increased steroid production considerably in both complete growth media (~4.4-fold) (Figure 4a) and serum-free media (~3.8-fold) (Figure 4b). In complete growth media, pregnenolone, progesterone and 17OH-progesterone increased, with the production of mineralocorticoids (*p* < 0.0001), together with the glucocorticoids (cortisol and cortisone) (*p* < 0.0001), also increasing. DHEA-S was the only androgen which increased considerably (*p* < 0.0001), with no additional increased androgen production. In serum-free media, 8BrcAMP stimulated the production of only 11-deoxycorticosterone and corticosterone (*p* < 0.0001), but levels remained more than ~2- and ~10-fold lower than stimulated levels in complete growth media, respectively. Glucocorticoid production was increased, together with androgen production (~2-fold), although their production was lower than in complete growth media.

The untargeted analysis of steroid production in H295A cells in both complete (Figure 4c) and serum-free media measured 11αOH-progesterone, 16OH-progesterone, pregnanetriol, 11OH-androstenedione and 11OH-testosterone. Minimal 11K-testosterone and 5α-reduced products were measured; however, the catalytic activity of 11βHSD2 was evident in the production of 11K-androstenedione. 8BrcAMP stimulated the production of pregnanetriol and 11K-androstenedione. Under serum-free conditions in the presence of metformin, most steroids showed a trend towards decreased levels and the ratio of progesterone/pregnenolone, denoting 3βHSD2 activity, decreased ~2-fold (Appendix A).

#### 2.1.3. Mouse Y-1 Cells

The Y-1 cell model was steroidogenic when cultured in complete growth media (Figure 5), predominantly producing pregnenolone and progesterone. Minimal mineralocorticoids, glucocorticoids and androgens were produced in this cell model. Although cortisol levels could be measured, these levels are indicative of the cortisol levels measured in the complete growth media (Appendix A). Serum-free conditions reduced the total steroid output by ~2-fold (Appendix A; however, progesterone (*p* ≤ 0.01), 11-deoxycorticosterone (*p* < 0.0001), DHEA (*p* ≤ 0.05) and androsterone (*p* ≤ 0.001) levels increased while corticosterone and cortisol levels decreased to trace amounts.

Stimulation with 8BrcAMP increased the steroid output ~3-fold (Figure 5a), predominantly increasing pregnenolone (~4-fold) and 11-deoxycorticosterone (*p* ≤ 0.01) levels, while progesterone was also increased ~6-fold. The untargeted analysis of steroid production in Y-1 cells in complete growth media (Figure 5b) measured the production of downstream progesterone metabolites, including pregnanolone, 5α-pregnanolone, 17α-hydroxypregnanolone, 6α-hydroxypregnanolone, together with pregnanetriol and 11-ketopregnanetriol. Progesterone metabolites with hydroxyl moieties at C6, C11, C16 and C20 were also measured. Upon 8BrcAMP stimulation, 16OH-progesterone, 6βOH-progesterone, 17α-hydroxypregnanolone and 6α-hydroxypregnanolone production increased. Although minimal steroid production was measured under serum-free conditions in the presence of metformin, the ratio of progesterone/pregnenolone, denoting 3βHSD2 activity, decreased ~1.7-fold (Appendix A).

#### 2.1.4. Mouse OS-3 Cells

The OS-3 cells are a mutant strain of Y-1 cells and were steroidogenic when cultured in complete growth media, predominantly producing pregnenolone and progesterone (Figure 6). Similarly to the Y-1 cell model, significant amounts of glucocorticoids or androgens were not produced in the OS-3 cell model. The cortisol levels that were measured were indicative of the cortisol levels measured in complete growth media (Appendix A). Serum-free conditions reduced the total steroid output by ~8-fold (Appendix A) and shifted the steroid profile. Progesterone (*p* < 0.0001), 11-deoxycorticosterone (*p* < 0.0001) and DHEA (*p* ≤ 0.01) levels were increased, with a concurrent decrease in pregnenolone (*p* < 0.0001) and corticosterone (*p* ≤ 0.05).

8BrcAMP stimulated the total steroid output significantly by ~27-fold, with significant increases in pregnenolone (*p* < 0.0001) and progesterone, 17OH-progesterone and 11-deoxycorticosterone (*p* < 0.0001) (Figure 6a). The untargeted analysis of steroid production in OS-3 cells in complete growth media (Figure 6b) measured multiple downstream progesterone metabolites and, upon 8BrcAMP stimulation, the production of the metabolites increased substantially. The increase in pregnenolone and progesterone clearly shunted into downstream metabolic pathways catalyzed by SRD5A, AKR1C1 and AKR1C2 enzymatic activities.

### 2.2. Placenta Cell Models

#### 2.2.1. Human BeWo Cells

BeWo cells were steroidogenic under complete growth media conditions (Figure 7), with reduced total steroid output in serum-free media (~1.9-fold). Pregnenolone and progesterone were the dominant steroids produced, and 11-deoxycorticosterone, cortisol, cortisone, DHEA and androsterone were all below 0.1 nmol/g (Figure 7a). Stimulation with 8BrcAMP increased pregnenolone and progesterone slightly (~1.4-fold), while all other steroid levels were unchanged. Mineralocorticoids, glucocorticoids and androgens were not major products in this cell model. Minimal amounts of downstream progesterone metabolites were also measured in the untargeted analysis (Figure 7b) and included 5α/β-dihydroprogesterone, 3β-pregnanolone, 6β-hydroxypregnanolone and pregnanetriol. 8BrcAMP stimulation did not alter steroid levels, nor did it change the steroid profile.

#### 2.2.2. Human JEG-3 Cells

JEG-3 cells were steroidogenic under complete growth media conditions (Figure 8), and similar to BeWo cells, serum-free conditions reduced the total steroid output to negligible levels (~1.8-fold). JEG-3 cells predominantly produced pregnenolone and progesterone, with negligible cortisol, cortisone and DHEA (Figure 8a). Stimulation with 8BrcAMP increased pregnenolone and progesterone significantly (*p* ≤ 0.001); however, all other steroids remained low or undetected, and mineralocorticoids, glucocorticoids and androgens unstimulated. The downstream progesterone metabolites, which included 16OH-progesterone, 6αOH-progesterone, 6βOH-progesterone, 20αOH-progesterone and 20βOH-progesterone, with 5α/β-dihydroprogesterone, were detected in the untargeted analysis of steroid production in cells with and without stimulation (Figure 8b). 8BrcAMP did not alter steroid levels or change the steroid profile, with 5α/β-dihydroprogesterone remaining the predominant metabolite measured.

### 2.3. Gonadal Cell Model—Mouse MA-10 Cells

MA-10 cells were exceptionally steroidogenic under complete growth media conditions and serum-free media conditions (Figure 9). In complete growth media, cells produced mainly pregnenolone, progesterone, DHEA-S and androsterone, with minimal mineralocorticoids (11-deoxycorticosterone and corticosterone) and negligible glucocorticoids (Figure 9a). Serum-free conditions reduced the total steroid output by ~3-fold and reduced most steroid metabolites to trace amounts (Figure 9b). Negligible aldosterone and dihydrotestosterone end-products were produced in the cells under tested conditions, while cortisol levels indicate cortisol in the culture media (Appendix A).

Stimulation with 8BrcAMP increased steroid production substantially in both complete growth media (>900-fold) (Figure 9a) and serum-free media (~185-fold) (Figure 9b). In complete growth media, the progestogens increased together with mineralocorticoids (11-deoxycorticosterone, corticosterone), glucocorticoids (11-deoxycortisol, 21-deoxycortisol and cortisone), and androgens (DHEA, androstenedione and testosterone). Androsterone and etiocholanolone were not quantified under stimulated conditions due to the excessive production of progestogens producing cross-talk peaks in their quantification channel. A similar profile was observed in serum-free media as cells stimulated with 8BrcAMP also produced high levels of progesterone, resulting in cross-talk. 8BrcAMP also increased mineralocorticoids (11-deoxycorticosterone, corticosterone), glucocorticoids (11-deoxycortisol, cortisol) and androgens (DHEA, androstenedione and testosterone) significantly.

The untargeted analysis of steroid production in MA-10 cells in both complete (Figure 9c) and serum-free media measured abundant downstream metabolites of progesterone, and of androstenedione and testosterone. The main androgen metabolites produced were androstenediol and 5α-androstanedione. The main progesterone metabolites produced were hydroxylated and reduced derivatives, most prominently 20αOH-progesterone, 5α-tetrahydroprogesterone and 5α/β-pregnane-3α, 20α-diol (20OHTHP). The stimulated production of progesterone resulted in significantly increased downstream progesterone metabolites that were very low in complete growth media without 8BrcAMP. Metabolites included 16OH-progesterone, 11αOH-progesterone and 11βOH-progesterone. The significantly increased androstenediol, androstenedione and testosterone resulting from 8BrcAMP stimulation yielded 5α-androstanedione, 11OH-androstenedione and 11OH-testosterone.

### 2.4. Non-Steroidogenic Cell Models

We additionally investigated steroid metabolism in human placental HTR-8/SVneo cells and in human ovary granulosa-like KGN and OVCAR-3 cells. These three cell models were non-steroidogenic in complete growth media and under serum-free conditions, and 8BrcAMP did not stimulate steroidogenesis (Table 1).

## 3. Discussion

This study explored the de novo biosynthesis of steroid hormones in adrenal, placental and gonadal immortalized cell models to establish comprehensive profiles. A comparative analysis of steroidogenesis was conducted in both human and mouse cell models. Pregnenolone biosynthesis and its subsequent metabolism was determined under basal, stimulated and serum-free conditions. The biosynthesis of pregnenolone from cholesterol is catalyzed by cytochrome P450 11A1 (side-chain cleavage, CYP11A1) and is the first and rate limiting step of steroidogenesis, supported by steroidogenic acute regulatory protein (StAR) [18]. Our findings confirm metabolic differences between steroidogenic and non-steroidogenic models, with data in support of the expression of *CYP11A1* and *StAR* (Appendix A), and all steroidogenic cell models showed evidence of endogenous CYP11A1 activity. Our comprehensive steroid profiles furthermore show clear metabolic differences between steroidogenic adrenal, placental and gonadal models catalyzed by steroidogenic enzymes [19,20,21,22,23,24].

Pregnenolone, which is readily converted by steroidogenic enzymes, is either shunted towards androgens via 17α-hydroxypregnenolone (17OH-pregnenolone) catalyzed by cytochrome P450 17A1 (17α-hydroxylase/17,20-lyase, CYP17A1) or serves as a 3βHSD substrate to form mineralocorticoids and glucocorticoids (Figure 1). 3βHSD catalyzed the conversion of pregnenolone to progesterone, observed in all cell models and decreased in adrenal cells treated with metformin. The influence of metformin on H295R cells has previously been described [17]. Our data additionally show the inhibition of 3βHSD activity in H295A and Y-1 cells and most prominently in OS-3 cells. Metformin also decreased the ratio of androstenedione/DHEA in H295R cells. Both pregnenolone and 17OH-pregnenolone, once converted by 3βHSD, lead to the production of mineralocorticoids and glucocorticoids. 17OH-pregnenolone was detected in the adrenal models except the mouse Y-1 cells, possibly due to the rapid conversion to downstream metabolites. Comparing the H295R and H295A cells, data show that the latter produced more metabolites in the mineralocorticoid pathway under basal conditions, which underscores the H295A cell model as an appropriate model for studies into mineralocorticoid biosynthesis. However, under stimulated conditions metabolites in both mineralocorticoid and glucocorticoid pathways were considerably higher in the H295R cells compared to the H295A cells. The H295R cells showed considerable glucocorticoid production, stimulated and unstimulated, the model thus more appropriate for investigations into glucocorticoid production. The choice of cell model would require careful consideration of experimental conditions to investigate modulated glucocorticoid production. The H295A cells produced considerably more cortisol upon stimulation, but upstream glucocorticoids (11-deoxycorticosterone, corticosterone and 11-deoxycortisol) were considerably higher in the H295R cells. The mouse adrenal models, in stark contrast to the human adrenal cells, produced negligible mineralocorticoids and glucocorticoids, with their de novo steroidogenesis predominantly yielding pregnenolone. Progesterone and an array of its downstream metabolites were detected in the mouse adrenal models, markedly different in metabolite profile and concentrations. The targeted steroid profiles in the BeWo and JEG-3 placenta cells were similar to the mouse adrenal models, and both pregnenolone and progesterone were also the predominant steroids. Both placental models, which are the most frequently used in in vitro studies have been reported to produce high levels of progesterone [25,26,27,28,29,30,31]. Our data showed that in JEG-3 cells, the progesterone levels were higher than pregnenolone but vice versa in the BeWo cells.

The MA-10 cell model (the only established immortalized Leydig-like cells available to date), on the other hand, was markedly different to the adrenal and placenta models. These cells are characterized by high de novo progesterone and testosterone production [32], and we found that progesterone was the dominant metabolite, even when cultured in serum-free media. In addition, the concentrations of the untargeted metabolites exceeded those of the targeted metabolites when cells were stimulated. The MA-10 cells and OS-3 mouse adrenal cells also produced 11-deoxycorticosterone (and corticosterone in MA-10 cells) upon stimulation with 8BrcAMP. Therefore, these cells do possess the capacity for producing these corticosteroids; however, these levels were negligible in serum-free media in OS-3 cells but persisted in MA-10 cells under serum-free-media-stimulated conditions. This upregulation of steroid production with the addition of 8BrcAMP is in line with protein kinase A/mitogen-activated protein kinase signaling regulating most steroidogenic enzymes [33].

Androgen production was limited to H295R, H295A and MA-10 cells, with unique hydroxylated and reduced metabolites measured in H295R cells. Androgen precursor formation from DHEA to downstream products could be traced in the H295R cell model. We also detected 5α-androstanedione and 11β-hydroxy-5α-androstanedione, the 5α-reduced forms of the two most prominent androgens produced in these cells, androstenedione and 11OH-androstenedione. 11OH-testosterone and 11K-testosterone were negligible, while the increased production of 11K-androstenedione upon 8BrcAMP stimulation highlights the conversion of 11OH-androstenedione catalyzed by endogenous 11βHSD2. The H295A cells, in contrast to H295R cells, produced more 11OH-androstenedione than androstenedione, owing to increased cytochrome P450 11B1 (11β-hydroxylase, CYP11B1) activity in H295A cells. The increased CYP11B1 activity observed is substantiated in the production of cortisol and corticosterone that were both higher than their precursors, 11-deoxycortisol and 11-deoxycorticosterone, respectively, in the H295A cells compared to the H295R cells. MA-10 cells mainly produced the classical androgens, androstenedione and testosterone, in comparison to the 11-oxy androgens. This has also been put forward in humans, in that the gonads do not contribute to circulatory 11-oxy androgen levels [34,35]. Mature Leydig cells primarily use the Δ5 pathway to biosynthesize testosterone [32,36], with the MA-10 cell model following this route to first produce DHEA, then androstenedione and finally testosterone upon 8BrcAMP stimulation. The Δ4 pathway to testosterone would also be active, as MA-10 cells produced substantial amounts of progesterone and 17OH-progesterone which would produce androstenedione and testosterone, as found in rodents [37].

Our data show that if the metabolism of pregnenolone and progesterone is not directed towards corticosteroids or androgens upon 8BrcAMP stimulation, steroid metabolism is directed towards downstream progesterone metabolites, most prominent in MA-10 cells. Stimulation increased the progesterone metabolic pathway considerably in MA-10 cells, and all metabolites significantly increased, resulting in hydroxylations at C6, C11, C17 and C20, as well as the reduction in the C4/5 double bond, and at C3. These conversions highlight endogenous expression of *Srd5a1* and *Akr1c14* [32,38], also observed in OS-3 cells and to a lesser extent in Y-1 cells. Marked progesterone downstream conversion was also observed in H295R cells, with the prominent CYP17A1 catalyzed production of 17OH-pregnenolone and 16OH-progesterone.

In comparing the placental cell models, the most interesting finding is that the HTR-8/SVneo cell model did not show de novo steroidogenesis, while the BeWo and JEG-3 cell models did, albeit limited to pregnenolone, progesterone and notable 5α/β-dihydroprogesterone. These results parallel what is known about placental steroidogenesis, and that the placenta requires precursor steroid metabolites provided by the maternal adrenal and fetal adrenal for abundant steroidogenesis [39]. As no downstream metabolites were detected in HTR-8/SVneo cells, together with ovary KGN and OVCAR-3 cells, we categorized these cells as non-steroidogenic. The steroidogenic enzymes required for the initiation of steroidogenesis might not be readily expressed in these cell models, or alternatively, these cells require steroid substrates to initiate downstream metabolism, leading to measurable steroid levels.

In this study, the investigation under serum-free media conditions was important and allowed the comparison with the standardized complete growth media for the growth and maintenance of the cell models. Our intention was to profile basal steroid production of cell models in the presence of complete growth media reflecting normal growth conditions. Serum is an important supplement for cell proliferation; however, they contain traces of steroid metabolites. These trace amounts of steroid metabolites could potentially influence the experimental outcome [40,41,42], and mass spectrometry can detect these trace amounts. This holds true in the current study, in which OVCAR-3 and OS-3 cells were grown in the presence of >15% serum supplemented culture media and cortisol was detected in the media of these cells, and in the complete growth media of MA-10 and Y-1 cells. The contribution of the trace amounts of cortisol in the media to the steroid profiles are also visible when comparing the profile in complete growth media with the profile in serum-free media. It is therefore important to distinguish between growth media and experimental media, if steroid measurements are the main experimental outcome when using these cell models. Hence, serum is usually charcoal-stripped of steroid hormones before use. In this regard, differences in steroid content between different batches of charcoal-stripped serum should also be taken into account as charcoal-stripping is not always absolute [43]. Mass spectrometry profiling of batches of complete growth media (plus serum) is therefore encouraged to standardize experimental data. Furthermore, as cell culture experiments and subsequent steroid extraction protocols inevitably have contact with plastic- and glassware, care should be taken in optimizing these protocols for steroid profiling, as certain steroid hormones might be retained on these surfaces [44].

The current study analyzed steroid metabolites using mass spectrometry, grouped into main steroid classes characterized by C_21_ and C_19_ steroid chemical backbones. However, a major steroid class which was not measured are the C_18_ estrogens. Estrogen metabolites are readily produced from androgens, depending on the expression of cytochrome P450 19A1 (aromatase) and would be most relevant in ovary and placental cell lines [12,39]. The production of estrogens would be largely dependent on steroid substrate administration, which was not the scope of the current study. Another important consideration is that the cell models used in the study are human or mouse clonal cell lines and therefore a direct link to healthy normal physiology should be made with caution. These models have, however, been well established to inform on human endocrinology and steroidogenesis [10,21]. The translational capacity between human and mouse cell models should also be noted as a constraint. Comparing mouse adrenal steroidogenesis against human adrenal steroidogenesis is not practical as the two species differ in their enzymatic content and steroidogenic capacity [45,46,47]. This is underscored in the distinctly different steroid profiles determined under all the tested conditions in the current study. Therefore, in vitro experiments conducted in mouse cell models cannot be translated to human cell models and a matching result should not be expected.

Stimulation studies are always a helpful tool to understand active metabolic pathways, and additional stimulation studies with forskolin and adrenocorticotropic hormone (ACTH) could be informative in future steroid profiling studies using human and mouse cell models [16]. Receptor expression in the cell models should, however, be considered, as ACTH stimulation in OS-3 cells would, for instance, not function as these cells do not express the melanocortin receptor 2 [48]. A relevant application of this study is steroid profiling in knockdown experiments when, for example, a co-enzyme such as cytochrome P450 oxidoreductase (POR), which is necessary for steroidogenesis, is knocked down in H295R cells. POR mutations can then be transiently transfected into these cells, the resulting altered steroid levels measured (stimulated/unstimulated) and compared to the wild-type cells. These experiments would inform the functionality of POR mutations found in POR-deficiency patients. Moreover, steroid profiling adrenal cell models in the presence of enzymatic inhibitors, such as CYP17A1 inhibitors, which targets excess adrenal androgen production, relevant to castration-resistant prostate cancer, is an ongoing application of this work in our group. Including all the steroid metabolites, steroid precursor accumulation and the metabolism of these steroids into other pathways can be efficiently tracked, pinpointing also off-target effects of these inhibitors. Lastly, co-culturing of placental and adrenal cells to better understand the interaction between the (fetal) adrenal and the placenta has been investigated [25,49], highlighting the current shift in research. A predominant focus on monolayer cell culturing is moving towards a more integrative 3D culturing. Combining cell types, together with the drive to utilize primary cell cultures, human organoids or induced pluripotent stem cells are now common practice to better understand human physiology [50,51,52]. In this regard, the expansion of the current study to model steroidogenesis using the cell lines in this study in a co-culturing environment would also be of interest.

In conclusion, we provide data on comprehensive steroid profiling including mineralocorticoid, glucocorticoid, progesterone and androgen steroid pathways in routinely used steroidogenic cell models. We detail not only the known classical steroid metabolic pathways in these cell models, but also alternative downstream pathway metabolites. The data confirm that these cell models present as experimental tools for molecular research related to active steroidogenic pathways, which could include the testing of novel enzymatic inhibitory or activating compounds or endocrine-disrupting chemicals and their effects on steroid hormones. In this study, we established standardized experimental cell culturing conditions and linked these to a characteristic steroid output, thereby providing a standardized protocol through which new cell models can be investigated. The data presented here would facilitate future investigations in utilizing these cell models with a clear understanding of their steroidogenic capacity.

## 4. Materials and Methods

### 4.1. Materials

Steroids, labeled internal standards and deuterated steroids, were purchased from Steraloids Inc. (Newport, RI, USA), Merck (Sigma-Aldrich, Darmstadt, Germany) and Cerilliant (Dorset, UK). Fetal bovine serum (FBS; 10270-106), horse serum (HS; 16050-122), penicillin-streptomycin (Pen-strep; 15140-122), L-glutamine (100×, 200 mM; 25030) and HEPES (1 M; 15630) were purchased from Gibco (Thermo Fisher Scientific, Waltham, MA, USA). Protease inhibitor cocktail tablets (1697498) were from Roche (Manheim, Germany) and NuI serum (355500) and ITS pre-mix (Universal cell culture supplement; 354351) were from BD Biosciences (Franklin Lakes, NJ, USA). Gelatine (G9391) and EDTA (0.015 g; 3610) were from Fluka (Charlotte, NC, USA) and Tris (0.122 g; 8382), Triton X-100 (T8787), 8BrcAMP (B5386; 25 mg), metformin-hydrochloride (catalog no. PHR1084, Lot no. LRAB3694) and zinc sulfate heptahydrate (Z4750) from Merck (Sigma-Aldrich, Darmstadt, Germany). Phosphate-buffered saline (PBS, pH 7.3) was provided by Inselgruppe Inselspital Apotheke Bern (13102084) and the protein determination kit was purchased from BioRad (DC Protein Assay) (Hercules, CA, USA). Formic acid, methanol and acetonitrile (all LC-MS grade) were from Biosolve (Valkenswaard, The Netherlands) and double charcoal-stripped, delipidized human serum was obtained from Golden West Diagnostics (Temecula, CA, USA). Analytical grade chemicals and tissue culture requirements were supplied by reliable scientific houses.

### 4.2. Cells

Cell models (Table 2) were obtained from the American Tissue Culture Collection (ATCC^®^, Manassas, VA, USA), the Department of Nephrology and Hypertension (Inselspital, Bern University Hospital, Bern, Switzerland) (BeWo and HTR-8/SVneo cells), Prof. Dr. med A Lauber-Biason (University of Fribourg, Fribourg, Switzerland) (KGN cells) and from Prof. W.L Miller (University of California, San Francisco, CA, USA) (H295A and OS-3 cells).

Cells were grown to 70–80% confluency before experiments were initiated. Cells were considered evenly distributed and healthy following microscopic inspection of wells. Human and mouse cells were grown at 37 °C, 5% CO_2_, 95% humidity, and heat-inactivated (manually) FBS and HS were used in all cell culture media (Table 3). The optimum amount of serum was added to the OVCAR-3 (20% FBS), OS-3 (20%: 15% HS and 5% FBS) and MA-10 cells (15% HS). Serum-free media contained only the media with the addition of 1% Pen-strep. Of note, ATCC^®^ does not advise on any use or concentration of antibiotics in their culture methods. When given a choice, the lowest % of serum was chosen for this study.

### 4.3. Steroid Metabolism

Cells were passaged for a minimum of three times from the freezer stock before experiments were initiated. Cell density under experimental conditions was as follows: 5 × 10^4^ cells/mL for KGN, JEG-3, OVCAR-3, Y-1 and OS-3 cells; 8 × 10^4^ cells/mL for HTR-8/SVneo cells; 1 × 10^5^ cells/mL for BeWo and MA-10 cells; and 1 × 10^6^ cells/mL for H295R and H295A cells. Duplicate independent experiments (cells started from different freezer stocks), in triplicate wells or more, were completed for all treatments for all steroidogenic cells. Metformin was prepared in deionized water [1 M in 500 µL], the solution was syringe-filtered (0.45 µm) and stored at −20 °C until use, and 8BrcAMP was dissolved in DMSO [100 mM stock; 25 mg in 612.6 µL DMSO] and stored at −20 °C until use. MA-10 and BeWo cells were grown on gelatine-treated cell culture dishes and split into gelatine-treated cell culture plates for experimentation (0.1%/g gelatine in 100 mL PBS, autoclaved and stored at 4 °C until use).

Treatment categories for all cell models were complete growth media (which includes serum) and serum-free media (termed ‘serum-free studies’). These two treatment categories were also expanded to include the addition of 8BrcAMP (and the control treatment of DMSO only) termed ‘stimulation studies’. The serum-free media treatment category for the adrenal cell models also included an experimental protocol with the addition of metformin.

Timeline for complete growth media studies: Cells were plated in six-well plates and incubated for 48 h to allow attachment. Thereafter, the media was replaced with fresh media for 24 h, after which the media was replaced once more, and following a 24 h incubation period, the media was aliquoted into glass test tubes and the cells harvested.

Timeline for serum-free studies: Cells were plated in six-well plates and incubated for 48 h to allow attachment. Thereafter, the media was replaced with fresh serum-free media for 24 h, after which the media was replaced with serum-free media once more, and following a 24 h incubation period, the media was aliquoted into glass test tubes and the cells harvested.

Timeline for stimulation studies: Cells were plated in six-well plates and incubated for 48 h to allow attachment. Thereafter, the media was replaced with either fresh complete growth media or serum-free media for 24 h, after which the media was replaced with the respective media with the addition of 8BrcAMP (0.5 mM/well, 10 µL), and following a 24 h incubation period, the media was aliquoted into glass test tubes and the cells harvested. The control plates, with the addition of DMSO, followed the exact same protocol as described above.

Timeline for metformin studies: Adrenal cells were plated in six-well plates and incubated for 48 h to allow attachment. Thereafter, the media was replaced with fresh serum-free media with or without the addition of metformin (10 mM/well, 20 µL) for 48 h, after which the media was aliquoted into glass test tubes and the cells harvested.

Protein determinations were completed following laboratory protocols and the standard assay protocol for Bio-Rad. Passive cell lysis buffer was prepared with EDTA (0.015 g), Tris (0.122 g) and NaCL (0.437 g) in deionized water, with the addition of 1% Triton X-100 in a total volume of 50 mL (stored at 4 °C until use). The cOmplete^TM^ Mini protease cocktail was also added to the passive cell lysis buffer right before the cell harvesting and storage. One pellet of the protease cocktail was dissolved in 400 µL deionized water and stored at −20 °C until use. Briefly, cells were harvested in 1× PBS and each well pooled in 1.5 mL tubes. The cells were collected by centrifugation and the PBS aspirated. Thereafter, 50 µL cell lysis buffer (24:1, lysis buffer/protease enzyme inhibitor cocktail mix) was added to each tube, vortexed for 5 min and stored at −20 °C until further use. Two freeze–thaw cycles were carried out prior to protein determination. A standard curve was generated using BSA in lysis buffer (50 mg/mL) ranging from 0.0875 to 5.6 mg/mL (a seven-point curve), in duplicate, which included a blank (lysis buffer only). Samples were pipetted into a 96-well transparent plate, 5 µL, followed by the addition of 25 µL reagent (reagent A + S; 100:2), and the subsequent addition of 200 µL reagent B. The plates were incubated at room temperature for 20 min, after which the absorbance was measured at 650 nm using the Softmax Programme 5.3 and the Bucher Biotec SpectraMax M2.

### 4.4. Steroid Extraction, Separation and Quantification

Reference standards and quality controls samples (100 µL) were prepared in 500 µL double charcoal-stripped, delipidized human serum, after which 38 µL of the isotopically labeled standards in methanol (3.8 nM each) were added to each sample. A protein precipitation step using zinc sulfate and methanol followed, and steroids were extracted using solid-phase extraction with an OasisPrime HLB 96-well plate (Waters Corporation, Milford, MA, USA). Experimental aliquots were centrifuged to remove cell debris and 1 mL was used for the analysis (without a protein precipitation step). After the addition of the labeled standards, steroids were extracted using solid-phase extraction, as described above and as previously published [5]. Samples were resuspended in 100 μL 33% methanol in water and 20 μL was injected into the LC-MS instrument (Vanquish UHPLC coupled to a QExactive Plus Orbitrap, Thermo Fisher Scientific, Waltham, MA, USA) using an Acquity UPLC HSS T3 column (Waters Corporation, Milford, MA, USA). Details of the LC-MS method, validation, calibrants, quality controls, internal standards, and lower limit of accurate quantifications (LLOQs) have been published previously [5]. The LLOQs and abbreviations are also shown in Appendix A.

### 4.5. Data Processing

Mass spectral data processing was completed using TraceFinder 4.0 (Thermo Fisher Scientific, Waltham, MA, USA). Steroid concentrations were normalized to total protein and data visualized using GraphPad Prism (version 10.6.0) and the results are shown as means ± SD. Steroid metabolites detected below the LLOQ are shown as not detected in the figures. Statistical analysis comparing two samples was performed on the natural log transformed data using either unpaired *t* tests with Welch correction or Mann–Whitney U tests on each row, and a two-stage linear step-up procedure of Benjamini, Krieger and Yekutieli (false discovery rate approach; Q = 1%). An indication of significance is shown as * *p* ≤ 0.05, ** *p* ≤ 0.01, *** *p* ≤ 0.001, **** *p* < 0.0001.

Additional data processing: Hydroxylated and reduced androgens and progestogens, termed ‘additional steroid profiling’, were measured in full MS mode only. Calibration curves were compared to androstenedione and progesterone, which allowed the generation of correction/response factors. Having established the response factors of all additional 34 steroids relative to steroids in the targeted workflow [5], we were also able to quantify these analytes. Our database therefore contains mass spectra and retention times obtained from authentic standards, and these parameters are then used to analyze the LC-MS data for these compounds quantified relative to either androstenedione or progesterone standard curves.

## Figures and Tables

**Figure 1 ijms-26-09721-f001:**
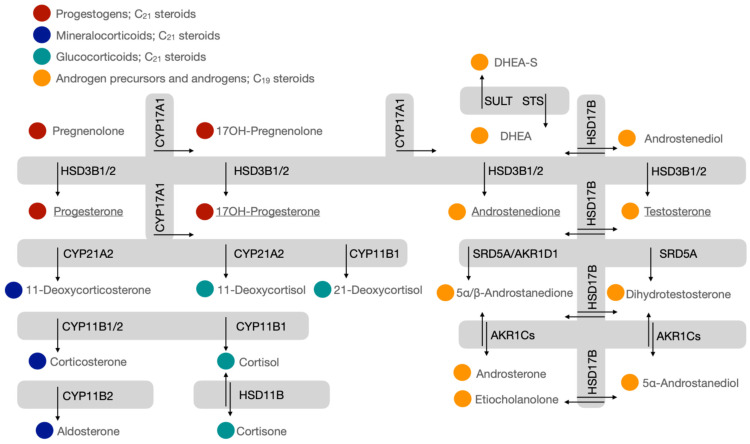
Steroid biosynthesis and metabolic pathways. **Steroids**: 17OH-Pregnenolone, 17α-hydroxypregnenolone; 17OH-Progesterone, 17α-hydroxyprogesterone; DHEA, dehydroepiandrosterone; DHEA-S, dehydroepiandrosterone sulfate. **Steroidogenic enzymes:** CYP17A1, cytochrome P450 17A1 (17α-hydroxylase/17,20-lyase); HSD3B1/2, 3β-hydroxysteroid dehydrogenase type 1/2; CYP21A2, cytochrome P450 21A2 (steroid 21-hydroxylase); SULT, sulfotransferase; STS, steroid sulfatase; HSD17B, 17β-hydroxysteroid dehydrogenase; SRD5A, steroid-5α-reductase (type 1 and 2); AKR1Cs, 3α-hydroxysteroid dehydrogenases; CYP11B1, cytochrome P450 11B1 (11β-hydroxylase); CYP11B2, cytochrome P450 11B2 (aldosterone synthase); HSD11B, 11β-hydroxysteroid dehydrogenase; and AKR1D1, 5β-reductase (aldo-keto reductase family 1 member D1). Underlined steroid metabolites; steroid substrates for steroidogenic pathways described in Figure 2.

**Figure 2 ijms-26-09721-f002:**
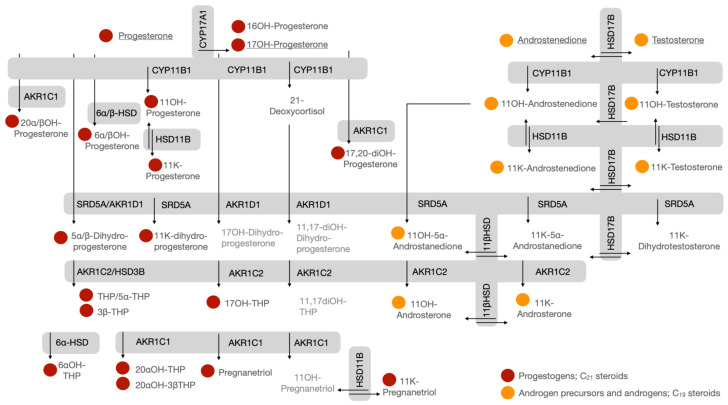
Additional downstream steroid metabolic pathways. **Steroids:** 16OH-Progesterone, 16α-hydroxyprogesterone; 20α/βOH-Progesterone, 20α/β-hydroxyprogesterone; 6α/βOH-Progesterone, 6α/β-hydroxyprogesterone; 11OH-Progesterone, 11α/β-hydroxyprogesterone; 11K-Progesterone, 11keto-progesterone; 17,20-diOH-Progesterone, 17α,20α-dihydroxyprogesterone; 11K-dihydroprogesterone, 11keto-dihydroprogesterone; THP, pregnanolone; 5α-THP, 5α-pregnanolone; 3β-THP, 3β-pregnanolone; 6αOH-THP, 6α-hydroxypregnanolone; 20αOH-THP, 20α-hydroxypregnanolone; 20αOH-3βTHP, 20α-hydroxy-3β-pregnanolone; 11OH-Pregnanetriol, 11β-hydroxypregnanetriol; 11K-Pregnanetriol, 11keto-pregnanetriol; 17OH-THP, 17α-hydroxypregnanelone; 11,17-diOH-THP, 11β,17α-dihydroxypregnanelone; 17OH-dihydroprogesterone, 17α-hydroxy-dihydroprogesterone; 11,17-diOH-dihydroprogesterone, 11β,17α-dihydroxy-dihydroprogesterone; 11OH-androstenedione, 11β-hydroxyandrostenedione; 11OH-testosterone, 11β-hydroxytestosterone; 11K-androstenedione, 11keto-androstenedione; 11K-testosterone, 11keto-testosterone; 11K-dihydrotestosterone, 11keto-dihydrotestosterone; 11K-5α-androstanedione, 11keto-5α-androstanedione; 11OH-5α-androstanedione, 11β-hydroxy-5α-androstanedione; 11OH-androsterone, 11β-hydroxyandrosterone; 11K-androsterone, 11keto-androsterone. Steroids in light gray are intermediate metabolites which were not included in our analytical method. **Steroidogenic enzymes:** CYP17A1, cytochrome P450 17A1 (17α-hydroxylase/17,20-lyase); HSD17B, 17β-hydroxysteroid dehydrogenase; SRD5A, steroid-5α-reductase (type 1 and 2); AKR1C2, 3α-hydroxysteroid dehydrogenase; CYP11B1, cytochrome P450 11B1 (11β-hydroxylase); HSD11B, 11β-hydroxysteroid dehydrogenase; AKR1D1, 5β-reductase (aldo-keto reductase family 1 member D1); AKR1C1, 20α-hydroxysteroid dehydrogenase; 6α/β-HSD, 6α/β-hydroxysteroid dehydrogenase; and HSD3B, 3β-hydroxysteroid dehydrogenase.

**Figure 3 ijms-26-09721-f003:**
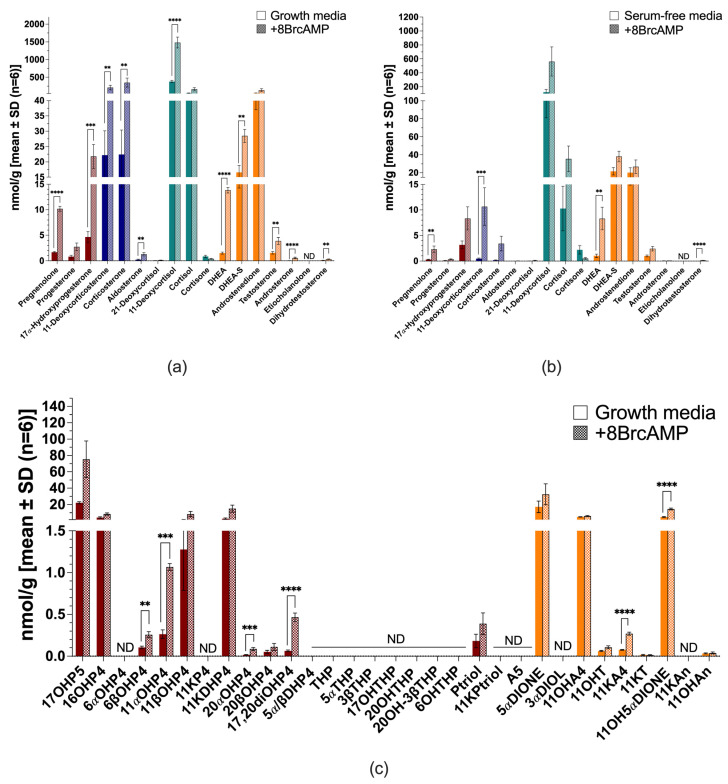
Steroid biosynthesis and metabolism in H295R cells. Targeted steroid analysis in complete growth media with (patterned bars) and without 8BrcAMP (**a**) and serum-free media with (patterned bars) and without 8BrcAMP (**b**); untargeted steroid analysis in complete growth media with (patterned bars) and without 8BrcAMP (**c**). Maroon bars, progestogens; blue bars, mineralocorticoids; turquoise bars, glucocorticoids; orange bars, androgens. ND not detected. ** *p* ≤ 0.01, *** *p* ≤ 0.001, **** *p* < 0.0001. Steroid abbreviations can be found in Appendix A.

**Figure 4 ijms-26-09721-f004:**
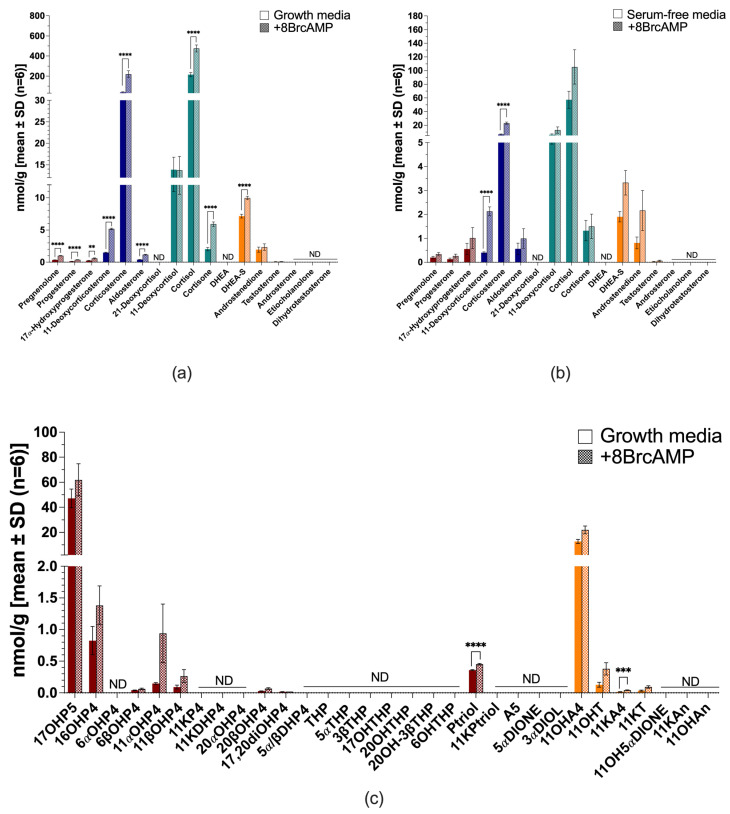
Steroid biosynthesis and metabolism in H295A cells. Targeted steroid analysis in complete growth media with (patterned bars) and without 8BrcAMP (**a**) and serum-free media with (patterned bars) and without 8BrcAMP (**b**); untargeted steroid analysis in complete growth media with (patterned bars) and without 8BrcAMP (**c**). Maroon bars, progestogens; blue bars, mineralocorticoids; turquoise bars, glucocorticoids; orange bars, androgens. ND, not detected. ** *p* ≤ 0.01, *** *p* ≤ 0.001, **** *p* < 0.0001. Steroid abbreviations can be found in Appendix A.

**Figure 5 ijms-26-09721-f005:**
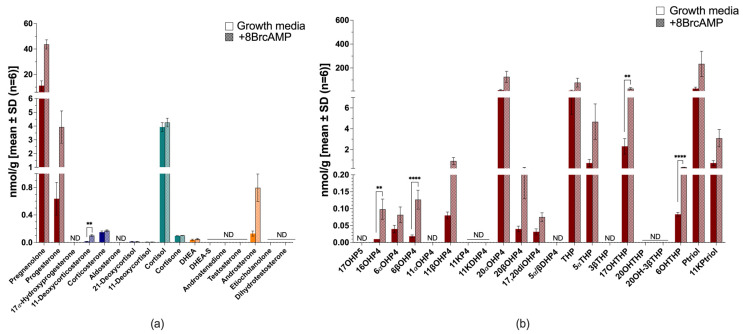
Steroid biosynthesis and metabolism in Y-1 cells. Targeted steroid analysis in complete growth media with (patterned bars) and without 8BrcAMP (**a**) and untargeted steroid analysis in complete growth media with (patterned bars) and without 8BrcAMP (**b**). Maroon bars, progestogens; blue bars, mineralocorticoids; turquoise bars, glucocorticoids; orange bars, androgens. ND, not detected. ** *p* ≤ 0.01, **** *p* < 0.0001. Steroid abbreviations can be found in Appendix A.

**Figure 6 ijms-26-09721-f006:**
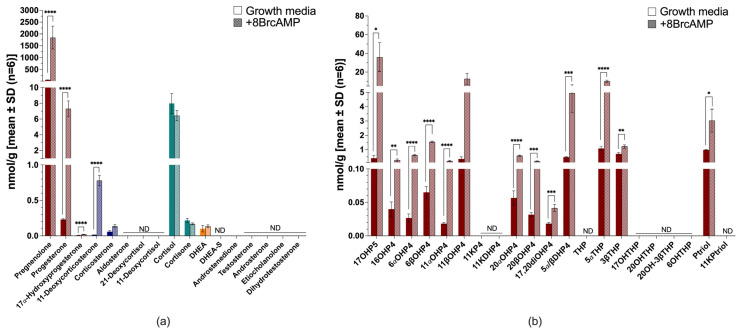
Steroid biosynthesis and metabolism in OS-3 cells. Targeted steroid analysis in complete growth media with (patterned bars) and without 8BrcAMP (**a**) and untargeted steroid analysis in complete growth media with (patterned bars) and without 8BrcAMP (**b**). Maroon bars, progestogens; blue bars, mineralocorticoids; turquoise bars, glucocorticoids; orange bars, androgens. ND, not detected. * *p* ≤ 0.05, ** *p* ≤ 0.01, *** *p* ≤ 0.001, **** *p* < 0.0001. Steroid abbreviations can be found in Appendix A.

**Figure 7 ijms-26-09721-f007:**
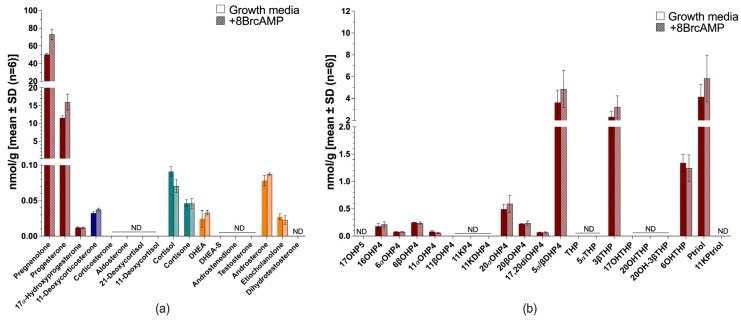
Steroid biosynthesis and metabolism in BeWo cells. Targeted steroid analysis in complete growth media with (patterned bars) and without 8BrcAMP (**a**) and untargeted steroid analysis in complete growth media with (patterned bars) and without 8BrcAMP (**b**). Maroon bars, progestogens; blue bars, mineralocorticoids; turquoise bars, glucocorticoids; orange bars, androgens. ND, not detected. Steroid abbreviations can be found in Appendix A.

**Figure 8 ijms-26-09721-f008:**
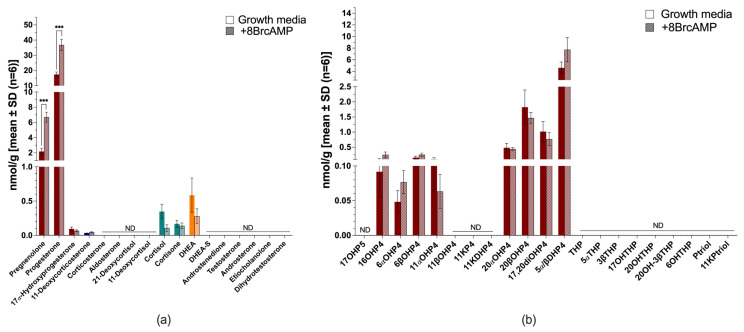
Steroid biosynthesis and metabolism in JEG-3 cells. Targeted steroid analysis in complete growth media with (patterned bars) and without 8BrcAMP (**a**) and untargeted steroid analysis in complete growth media with (patterned) and without 8BrcAMP (**b**). Maroon bars, progestogens; blue bars, mineralocorticoids; turquoise bars, glucocorticoids; orange bars, androgens. ND, not detected. *** *p* ≤ 0.001. Steroid abbreviations can be found in Appendix A.

**Figure 9 ijms-26-09721-f009:**
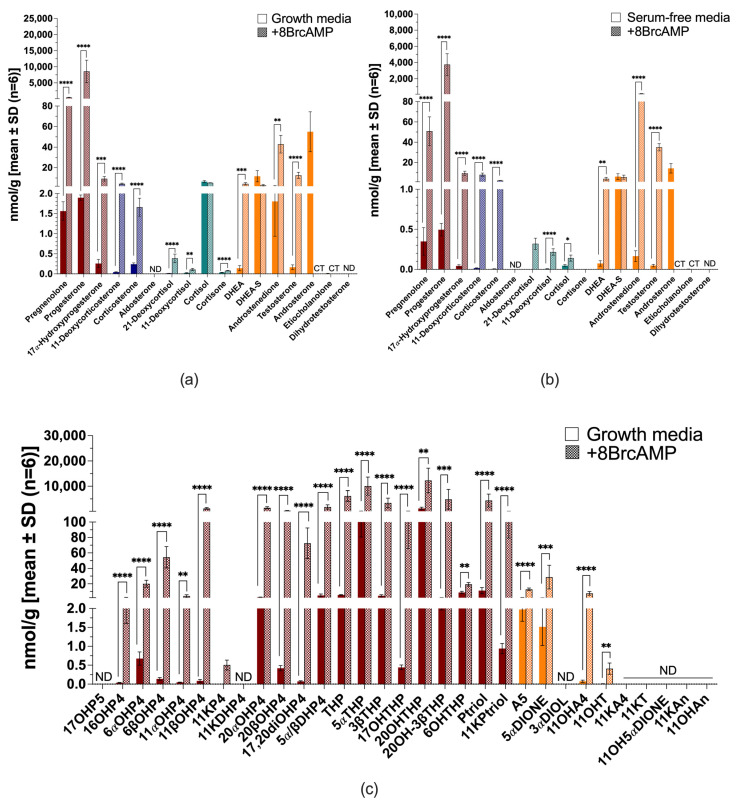
Steroid biosynthesis and metabolism in MA-10 cells. Targeted steroid analysis in complete growth media with (patterned bars) and without 8BrcAMP (**a**) and serum-free media with (patterned bars) and without 8BrcAMP (**b**); untargeted steroid analysis in complete growth media with (patterned bars) and without 8BrcAMP (**c**). Maroon bars, progestogens; blue bars, mineralocorticoids; turquoise bars, glucocorticoids; orange bars, androgens. ND, not detected; CT, cross-talk. * *p* ≤ 0.05, ** *p* ≤ 0.01, *** *p* ≤ 0.001, **** *p* < 0.0001. Steroid abbreviations can be found in Appendix A.

**Table 1 ijms-26-09721-t001:** Classification of steroidogenicity and the predominant steroid class biosynthesized under complete growth media conditions of the cell models included in the study. The larger the graphical representation of the steroid class produced in each cell model, the more this class is produced after a 24 h incubation period (
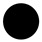
 ~300–500 nmol/g; 
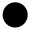
 ~200–300 nmol/g; 
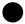
 ~60–100 nmol/g; 
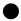
 ~40–60 nmol/g; 
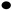
 ~5–20 nmol/g; 

 ~0–5 nmol/g). Maroon circles, progestogens; blue circles, mineralocorticoids; turquoise circles, glucocorticoids; orange circles, androgens.

Tissue of Origin	Cell Models	Classification	Steroid Class	Total Steroid Output (nmol/g)
Non-Steroidogenic	Steroidogenic	Non-Steroidogenic	Steroidogenic	Progestogens	Mineralocorticoids	Glucocorticoids	Androgens
+Serum	−Serum	−8BrcAMP	+8BrcAMP
Adrenal	H295R		✓		✓	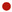	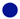	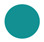	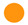	611.8	2586.6
H295A		✓		✓		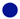	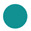	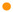	342.6	823.1
Y-1		✓		✓	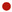				71.5	528.3
OS-3		✓		✓	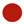		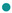		73.5	1934.7
Gonadal	KGN	✓		✓						0.2	0.2
OVCAR-3	✓		✓						0.9	0.8
MA-10		✓		✓			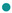	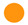	1475.5	71,486.9
Placenta	BeWo		✓		✓ ↓	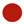				75.0	106.0
JEG-3		✓		✓ ↓	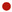				29.1	55.2
HTR-8/SVneo	✓		✓						0.0	0.0

**Table 2 ijms-26-09721-t002:** Characteristics of the cell models included in the study.

Tissue of Origin	Specie	Cell Line	Cell Type	Normal/Diseased
**Adrenal**	Human	NCI: H295R (ATCC^®^CRL-2128^TM^)	Epithelial	Carcinoma
		H295A	Epithelial	Carcinoma
	Mouse	OS-3	Epithelial	Carcinoma
		Y-1 (ATCC^®^CCL-79^TM^)	Epithelial	Carcinoma
**Gonadal**	Human ovary	NIH: OVCAR-3 (ATCC^®^HTB-161^TM^)	Epithelial	Adenocarcinoma
		KGN	Granulosa-like	Carcinoma
	Mouse testicular (Leydig)	MA-10 (ATCC^®^CRL-3050^TM^)	Epithelial-like	Carcinoma
**Placental**	Human	JEG-3 (ATCC^®^HTB-36^TM^)	Epithelial	Choriocarcinoma
		HTR-8/SVneo (ATCC^®^CRL-3271^TM^)	Epithelial	Normal
		BeWo (ATCC^®^CCL-98^TM^)	Epithelial trophoblast-like	Choriocarcinoma

**Table 3 ijms-26-09721-t003:** Culture media of the cell models included in the study.

Cell Line [Passage Number From Stock 1st; Passage Number From Stock 2nd]	Culture Media (Total Volume: 500 mL)
H295R [17/18; 24]	DMEM/F12 (Gibco 31330-038; containing 2.5 mM L-glutamine, 15 mM HEPES, 17.5 mM dextrose, 8.1 mg/L phenol red, 0.5 mM sodium pyruvate and 1200 mg/L NaHCO_3_) +NuI-serum I (5%; 25 mL) +Pen-strep (1%; 5 mL) +ITS pre-mix (0.1%; 0.5 mL)
H295A [25; 25]	RPMI-1640 (Gibco 21875-034; containing 2.1 mM L-glutamine, 11.1 mM dextrose, 5 mg/L phenol red and 2000 mg/L NaHCO_3_) +FBS (2%; 10 mL) +ITS pre-mix (0.1%; 0.5 mL) +Pen-strep (1%; 5 mL)
OS-3 [10; 10]	Ham’s F10 (Gibco 31550-023; containing 1 mM L-glutamine, 1.2 mg/L phenol red, 6.1 mM dextrose, 1 mM sodium pyruvate and 1200 mg/L NaHCO_3_) +HS (15%; 75 mL) +FBS (5%; 25 mL) +Pen-strep (1%; 5 mL)
Y-1 [17; 18]	DMEM/F12 (Gibco 11320-033; containing 2.5 mM L-glutamine, 8.1 mg/L phenol red, 17.5 mM dextrose and 2438 mg/L NaHCO_3_) +HS (7.5%; 37.5 mL) +FBS (2.5%; 12.5 mL) +Pen-strep (1%; 5 mL)
OVCAR-3 [7/10]	RPMI-1640 (Gibco 21875-034; containing 2 mM L-glutamine, 5 mg/L phenol red, 11.1 mM dextrose and 2000 mg/L NaHCO_3_) +FBS (20%; 100 mL) +Pen-strep (1%; 5 mL) +ITS (0.1%; 0.5 mL)
KGN [7/9]	DMEM/F12 (Gibco 21331-020; containing no L-glutamine, 8.1 mg/L phenol red, 17.5 mM dextrose, 0.5 mM sodium pyruvate and 2438 mg/L NaHCO_3_) +FBS (10%; 50 mL) +Pen-strep (1%, 5 mL) +HEPES (15 mM; 7.5 mL)
MA-10 [8; 14]	DMEM/F12 (Gibco 31330-038; containing 2.5 mM L-glutamine, 8.1 mg/L phenol red, 17.5 mM dextrose, 0.5 mM sodium pyruvate, 15 mM HEPES and 1200 mg/L NaHCO_3_) +HS (15%; 75 mL) +HEPES (20 mM; 10 mL) +Pen-strep (1%; 5 mL)
JEG-3 [44/48; 36]	EMEM (Gibco 21090-022; containing 10 mg/L phenol red, 5.6 mM dextrose and 2200 mg/L NaHCO_3_) +FBS (10%; 50 mL) +Pen-strep (1%; 5 mL) +L-glutamine (1%; 5 mL [2 mM])
HTR-8/SVneo [89; 92]	RPMI-1640 (Gibco 21875-034; containing 2.1 mM L-glutamine, 11.1 mM dextrose, 5 mg/L phenol red and 2000 mg/L NaHCO_3_) +FBS (10%; 50 mL) +Pen-strep (1%; 5 mL)
BeWo [26/27; 29]	DMEM/F12 (Gibco 11320-033; containing 2.5 mM L-glutamine, 8.1 mg/L phenol red, 17.5 mM dextrose and 2438 mg/L NaHCO_3_) +FBS (10%; 50 mL) +Pen-strep (1%; 5 mL)

## Data Availability

The original contributions presented in this study are included in the article/Appendix A. Further inquiries can be directed to the corresponding author.

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
