# Peer review of "Characterization of Steroid Metabolic Pathways in Established Human and Mouse Cell Models"

_ijms, 2025, doi:10.3390/ijms26199721_

Round 1
Reviewer 1 Report
Comments and Suggestions for Authors
Characterization of steroid metabolic pathways in established
human and mouse cell models (ijms-3877451)
Therina du Toit and co-worker reported on steroidogenesis in human adrenal H295R and H295A; 17 placental BeWo and JEG-3; mouse Leydig MA-10; and mouse adrenal Y-1 and OS-3, cells. Besides Profiling of classic steroid metabolites, they focused also on downstream untargeted metabolites which are often not reported. Furthermore, they conducted their analyses with respect to different culturing conditions (basal, stimulated and serum-free) using liquid chromatography-mass spectrometry. The manuscript comprises detailed protocols and provides steroid profiles to support researchers for further in vitro investigations. Therefore, the here presented work is very helpful in experimental research on steroidogenesis.
The manuscript is well written and all necessary information are included. Furthermore, they have summarized their results comparing different cell types and conditions table 1. The concentrations of different metabolites are particularly helpful for others with respect to pharmacological interventions and accompanying statistics.
I have no further comments and I congratulate the authors on their valuable manuscript.
Reviewer 2 Report
Comments and Suggestions for Authors
The manuscript titled “Characterization of steroid metabolic pathways in established human and mouse cell models” effectively highlights the importance of steroid hormones through its experimental work. Overall, the topic holds merit, though it needs some improvements before possible publication in the journal.
- For greater scientific rigor and precision, please specify examples of the cell models (in line 111) referenced, thereby enhancing clarity and contextual depth.
- The term ‘de novo’ should be consistently italicized throughout the manuscript to conform to standard scientific writing.
- The concluding statement on future directions is appropriate, but could be more impactful by suggesting 1–2 concrete research directions or applications that this work enables.
- A dedicated limitations section would strengthen the study, particularly addressing translational constraints from interspecies differences in steroidogenic enzyme expression.
Round 2
Reviewer 2 Report
Comments and Suggestions for Authors
None